

# A combined test for feature selection on sparse metaproteomics data—an alternative to missing value imputation

Sandra Plancade[1,*], Magali Berland[2,*], Mélisande Blein-Nicolas[3,4], Olivier Langella[3,4], Ariane Bassignani[2,4] and Catherine Juste[5]

[1] UR875 MIAT, Université fédérale de Toulouse, INRAE, Castanet-Tolosan, France
[2] Université Paris-Saclay, INRAE, MGP, Jouy en Josas, France
[3] Université Paris-Saclay, CNRS, INRAE, AgroParisTech, GQE-Le Moulon, Gif-sur-Yvette, France
[4] Université Paris-Saclay, CNRS, INRAE, AgroParisTech, PAPPSO, Gif-sur-Yvette, France
[5] Micalis Institute, Université Paris-Saclay, INRAE, AgroParis Tech, Jouy-en-Josas, France
* These authors contributed equally to this work.

## ABSTRACT

One of the difficulties encountered in the statistical analysis of metaproteomics data is the high proportion of missing values, which are usually treated by imputation. Nevertheless, imputation methods are based on restrictive assumptions regarding missingness mechanisms, namely "at random" or "not at random". To circumvent these limitations in the context of feature selection in a multi-class comparison, we propose a univariate selection method that combines a test of association between missingness and classes, and a test for difference of observed intensities between classes. This approach implicitly handles both missingness mechanisms.

We performed a quantitative and qualitative comparison of our procedure with imputation-based feature selection methods on two experimental data sets, as well as simulated data with various scenarios regarding the missingness mechanisms and the nature of the difference of expression (differential intensity or differential presence). Whereas we observed similar performances in terms of prediction on the experimental data set, the feature ranking and selection from various imputation-based methods were strongly divergent. We showed that the combined test reaches a compromise by correlating reasonably with other methods, and remains efficient in all simulated scenarios unlike imputation-based feature selection methods.

# INTRODUCTION

Metaproteomics refers to the study of all proteins present in an ecosystem (soil, water, gut,…) at a given time. It allows for the qualitative and quantitative profiling of the tremendous diversity of proteins in complex biological samples. It is the method of choice to learn about which microorganisms are doing what in a microbial ecosystem. Therefore, metaproteomics moves beyond the genetic potential addressed by metagenomics, and it is generating rising interest and new international initiatives (https://metaproteomics.org/). Yet metaproteomics has long lagged behind metagenomics due to the lack of appropriate

Corresponding author
Sandra Plancade,
sandra.plancade@inrae.fr

tools, but impressive progress in LC-MS/MS technologies (liquid chromatography coupled with tandem mass spectrometry) makes it possible to decipher metaproteomes in a deep, broad and high throughput manner. However, processing of metaproteomics data is much less developed than for metagenomics, and statistical approaches developed for proteomics of single organisms cannot necessarily be transposed to complex ecosystems. Indeed, metaproteomics data are characterized by a huge diversity and specificity within and between samples; this generates large and sparse matrices of protein abundances which require dedicated analytical methods. In particular, selecting metaproteomic features that are shared by homogeneous clinical groups could facilitate the diagnosis or prognosis of a disease.

Feature selection methods (FSMs) can be classified in two categories. Wrapper and embedded methods make use of a classifier to select a set of features based on their discrimination ability, either with a recursive selection (wrapper) or by including a filtering into the classifier (embedded) (*Saeys, Inza & Larranaga, 2007*). While these methods enable the extraction of a reduced list of predictors, they are pointed out as potentially generating overfit (*Saeys, Inza & Larranaga, 2007*), and lead to the elimination of correlated features which may be detrimental to biological intepretation. In univariate methods, features are examined separately. These methods do not account for potential interactions amongst variables, but they enable the inclusion of more complex designs (batch effects, multiple effects, censoring,…).

One of the difficulties in implementing FSMs on shotgun proteomics and metaproteomics is handling the missing data. Indeed, LC-MS/MS technologies are known to generate a high rate of missing values and this phenomenon is enhanced in metaproteomics. On one hand, microbiota composition is largely specific to individuals, leading to a significant proportion of truly missing proteins. On the other hand, the high complexity of microbiota samples makes data acquisition and pre-processing particularly sensitive, and generates a higher technical variability than observed on proteomics data, leading to important measurement errors as well as missing values. The processes leading to missingness are diverse and may originate from any step of the pipeline, either biochemical, analytical or bioinformatics (*Lazar et al., 2016*). These mechanisms can be analysed in the framework developed by *Rubin (1976)*, who distinguishes Missing At Random (MAR) in which the probability for a feature to be missing is independent of its true abundance, and Missing Not At Random (MNAR) in which missingness depends on the abundance, including notably thresholding due to device detection limit. It is commonly recognized that both MAR and MNAR occur with LC-MS/MS technologies (*O'Brien et al., 2018*; *Lazar et al., 2016*), but neither the proportion of each mechanism on a data set nor the precise mechanism at the origin of a given missing value are known *a priori*.

Methods to address missing data in proteomics mostly rely on either missing value imputation (*Wang et al., 2020*) or statistical modelling of censoring mechanisms (*Karpievitch et al., 2009*; *Luo et al., 2009*, *O'Brien et al., 2018*), even if a few alternatives have arisen. Borrowing from both above mentioned categories, *Berg et al. (2019)* recently proposed a multiple imputation approach based on a MAR/MNAR model. *Gianetto et al.*

*(2020)* (R package `imp4p`) developed a statistical model that combines MAR and MNAR missing value imputation. Besides, *Webb-Robertson et al. (2010)* developed a filtering approach that circumvents missing values imputation by means of two successive filterings based on difference in terms of peptide occurrence, and difference in intensities among the non-missing observations. To the best of our knowledge, in the metaproteomics context, the treatment of missing values mostly relies on imputation (*Tang et al., 2020b*). A large number of imputation methods for proteomics or metaproteomics have been proposed in the literature (R package `NAguideR`, *Jin et al. (2021)*), and can be classified in three categories: (i) single value imputation, where missing intensities are replaced by the same value for all samples; (ii) global structure methods, in which imputation is based on correlations between the whole set of observations; (iii) local similarity imputation, based only on the most similar features.

In this article, we propose an approach which circumvents the limitations of missing value imputation and implicitly handles both MAR and MNAR mechanisms. This univariate feature selection method combines a presence/absence test which detects if the frequence of missingness is different between classes, and a test of the difference in observed intensities between classes, embedded in a permutation test procedure. We compared our method with three imputation-based FSMs, on two metaproteomics data sets: the first one from human gut microbiota of a cohort of coronary artery disease patients, and the second one from gut microbiota samples of pigs repeatedly measured in a diet perturbation experiment. Moreover, we made use of a set of technical replicates to explore missingness mechanisms.

## MATERIAL AND METHODS

### Experimental data sets

#### ProteoCardis

We used a subset of the data set generated in the ProteoCardis project, an association study between the human intestinal metaproteome and cardiovascular diseases, using fecal samples collected in the framework of the FP7 MetaCardis. (*Bassignani, 2019*; Section 1.6). Two classes were considered: patients with acute cardiovascular disease ($N = 49$) and healthy controls ($N = 50$). For each of these 99 subjects, the extracted gut microbiota was fractionated into its cytosolic and envelope compartments, which were analysed separately for their metaproteome, giving a total of 198 metaproteomes. Details on metaproteomics analyses can be found in *Bassignani (2019)* (Section 4.1.2). The cytosolic and envelope data sets are denoted by *ProteoCardis-cyto* and *ProteoCardis-env*. In order to investigate technical variability, eight biological samples from the ProteoCardis cohort were replicated seven times each, for both their cytosolic and envelope compartment analyses.

The peptides and the proteins they come from were identified using an original iterative method described in *Bassignani et al. (2021)*. Indistinguishable proteins, *i.e.*, those identified with a same set of peptides, were grouped into metaproteins (or protein subroups) using the parsimonious grouping algorithm of X!TandemPipeline (*Langella et al., 2017*). To simplify the writing, those protein assemblages are denoted "proteins" in

the following (*Bassignani et al., 2021*; *Bassignani, 2019*). Finally, intensities of proteins were calculated as the sum of the extracted ion currents of their specific peptides, using MassChroQ (*Valot et al., 2011*). Data are available at https://doi.org/10.15454/ZSREJA.

### Pigs

The data set Pigs (*Tilocca et al., 2017*) consists in fecal microbiota analyses on 12 pigs observed at six time points during a 4-week diet. Samples from weeks one and two (one observation per week) were gathered into a metabolic period, and samples from weeks three and four (two observations per week) into an equilibrium period. These two periods represent our classes of interest for this analysis, similarly to *Tang et al. (2020b)*. All details can be found in *Tilocca et al. (2017)*, and the data are available at ProteomeXchange PXD006224.

### Filtering of sparse features

FSMs were applied on log-transformed data after filtering out proteins with less than $\tau$ non-missing values, with $\tau$ equal to 40% of the size of the smallest class ($\tau = 20$ for *ProteoCardis* and 10 for *Pigs*). This value represents a compromise between a high threshold that may lead to the deletion of a large part of the features, and a low threshold where too little information would be available for some variables. Nevertheless, as the impact of the missing value treatment may depend on the proportion of missing values, complementary analyses were performed with higher threshold values (30, 40 and 50 for *ProteoCardis* data sets; 20 and 30 for *Pigs*).

### Statistical characteristics of the experimental data sets

Even after filtering of sparse features, *ProteoCardis* data sets are still highly sparse, with most proteins having more than half missing values while *Pigs* displays a larger proportion of proteins with very few missing values (Fig. S1, top). These differences of sparsity may originate from a higher similarity in terms of genetic background and diet among experimental animals. Moreover, many more proteins are significantly different between the two classes in *Pigs* than in *ProteoCardis* for all FSMs (Fig. S1, bottom). Thus *Proteocardis* and *Pigs* display different statistical characteristics, which enhance the robustness of the FSM comparison carried out in this paper.

## Analysis of replicates

Consider a technical replication of the analysis of a biological sample *i*. The probability that a feature *j* is missing in the technical replicate, given that the observed intensity $Y_{i,j}$ is equal to *y* in the original analysis, is defined as follows:

$$p_{i,j}^0(x) = \mathbb{P}(Y_{i,j}' = \text{NA}|Y_{i,j} = y)$$

with $Y_{i,j}'$ the observed intensity in the replicate. The replicate data sets were used to infer the missingness probability function under the assumption that the probability $p_{i,j}^0(x)$ is independent of the biological sample and of the feature: $p_{i,j}^0(x) = p^0(x)$. Then, observed intensities were stratified in 5% quantiles: $(y_0, \ldots, y_{20})$ and $p^0$ was approximated by:

$$p\left(\frac{x_\ell + x_{\ell+1}}{2}\right) = \mathbb{P}(Y'_{i,j} = NA | Y_{i,j} = [x_\ell, y_{\ell+1})) = \frac{\mathbb{P}(Y'_{i,j} = NA, Y_{i,j} \in [y_\ell, y_{\ell+1}))}{\mathbb{P}(Y_{i,j} \in [y_\ell, y_{\ell+1}))}$$

which was estimated by its empirical counterpart:

$$\frac{\sum_{j=1}^{J} \sum_{i=1}^{8} \sum_{r,r'=1,\ldots,7, r \neq r'} 1\!\mathrm{I}_{Y^r_{i,j}=x, Y'^{r'}_{i,j}=NA}}{J \times 8 \times 7 \times 6} \times \frac{J \times 8 \times 7}{\sum_{j=1}^{J} \sum_{i=1}^{8} \sum_{r=1,\ldots,7} 1\!\mathrm{I}_{Y^r_{i,j}=x}}$$

with $Y^r_{i,j}$ the intensity of the protein $j$ in the replicate $r$ of the biological sample $i$.

## Simulation study

A simulation study was conducted to illustrate the impact of both the nature of the biological difference between classes and the missingness mechanism. A realistic full data set was generated by kNN imputation of missing values on the log-transformed intensities from the *ProteoCardis-cyto* data set, after filtering of proteins with less than 10 non-missing values. Two classes of size 49 and 50 were randomly sampled among the 99 samples, so that no proteins are differentially expressed between the two classes except by chance. Then, difference of expression between classes was generated. First of all, 2,000 proteins were randomly selected to be truly differentially expressed, assuming either a difference in intensity (fold change), or a difference in probability of presence. Then, missing values were picked up assuming either MAR or MNAR mechanism, followed by filtering of proteins with less than 20 non-missing values. Details can be found in the Supplemental Material.

## Combined test

We propose a protein level univariate combined test that accounts for both missing and non-missing data. Consider $m$ features (here, proteins) observed in $n$ samples belonging to two or more classes. For each feature $j$, the difference of intensity between the two classes on non-missing observations, and the association between class and missingness are tested *via* the following linear mixed model (lmm) and generalised linear mixed model (glmm):

$$\begin{cases} Y_{i,j} = X_i\beta_j + Z_i u_j + \varepsilon_{i,j}, & i = 1, \ldots, n \text{ such that } Y_{i,j} \neq NA & (\text{mod} - \text{lmm}) \\ \log\dfrac{\mathbb{P}[Y_{i,j} \neq NA]}{\mathbb{P}[Y_{i,j} = NA]} = X'_i\beta'_j + Z'_i u'_j + \varepsilon'_{i,j} & i = 1, \ldots, n & (\text{mod} - \text{glmm}) \end{cases}$$

with $Y_{i,j}$ the log-transformed observed intensity of feature $j$ in sample $i$, $(X, X')$ design matrices of fixed effect including the class effect, $(Z, Z')$ design matrices of random effects, $(\varepsilon_{i,j})_i$ and $(\varepsilon'_{i,j})_i$ i.i.d. centered gaussian. Let $p_1$ and $p_2$ be the $p$-values of the F-test for class effect in models (mod-lmm) and (mod-glmm) respectively.

Let $S$ be the Fisher combined statistic defined as:

$$S = -\frac{1}{2}(\log p_1 + \log p_2). \tag{1}$$

$S$ is large if at least one of the two $p$-values $p_1$ and $p_2$ is small; moreover, if only one of the two $p$-values is small, $S$ is weakly affected by the value of the largest one. If the statistics of the two tests were independent, $S$ would be $\chi^2$-distributed under the global null hypothesis, but this assumption may be violated, especially under MNAR assumption, since low protein abundances could simultaneously lead to low observed intensities and high probability to be missing. Therefore, the distribution under the global null hypothesis is obtained by repeated permutations ($N^{perm}$) of the classes. In order to reduce the computing time and to increase the number of distinct values that can be taken by $S$, the distribution under the null hypothesis is assumed to be identical for all variables with the same proportion of missing values. Mathematical details are provided in the Supplemental Material.

### Design for ProteoCardis and Pigs data sets

For the *ProteoCardis* data sets, no random effect was considered and

$$X_i \beta_j = \beta_{j,0} + \beta_{j,1} 1\!I_{i \in \mathscr{C}_1} + \beta_{j,2} 1\!I_{i \in \mathscr{C}_2}$$

with $\mathscr{C}_1$ and $\mathscr{C}_2$ the two classes. For the *Pigs* data set, a random animal effect was added:

$$X_i \beta_j + Z'_i u'_j = \beta_{j,0} + \beta_{j,1} 1\!I_{i \in \mathscr{C}_1} + \beta_{j,2} 1\!I_{i \in \mathscr{C}_2} + u_{j,a(i)}$$

with $a(i)$ the animal on which sample $i$ was collected. The comparison between classes is performed based on the contrast $\beta_{j,1} - \beta_{j,2}$.

### Permutations framework

In the case of a complex design, single shuffling may be inappropriate since data are not freely exchangeable under the null hypothesis. Thus for *Pigs*, permutations were implemented such that the number of observations on each animal over each period was preserved (two time points and four time points over the first and second period respectively), using the R package `permute`.

For the implementation on the whole data set, we considered $N^{perm} = 10^5$, and for the prediction accuracy and the replicability on independent subsets, $N^{perm} = 10^4$. A larger number of permutations leads to a better precision of the $p$-values (of order $1/N^{perm}$) but at the cost of a larger computation time which increases linearly with $N^{perm}$. Note that the procedure can be parallelised very easily, since the distribution under the null hypothesis is computed separately for each possible number of missing values. Besides, the combined test requires a sufficient sample size, so that enough permutations can be realised to compute the statistic distribution under the null hypothesis with a good precision.

## FSMs based on NA imputation methods

The combined test was compared with feature selection procedures based on missing value imputation. We considered the three imputation methods proposed by *Tang et al. (2020a)* (package `metaFS`), namely: (i) single value imputation, where all missing value are replaced by the smallest intensity observed in the data set; (ii) k-nearest neighbours (kNN); (iii) singular value decomposition (SVD). Following *Wang et al. (2020)*, kNN was implemented using the R function `SeqKNN` with a number of neighbours $k = 10$ and SVD was implemented using the function `pca` (package `pcaMethods`) with two components. Then, the linear mixed model (mod-lmm) was applied on the vectors of observed and imputed intensities. The choice of a lmm testing procedure after imputation guarantees that the differences observed between the imputation-based FSMs and the combined test are exclusively due to the treatment of missing values. The imputation-based FSMs are denoted as follows.

- **Single-lmm:** log-transformation + single value imputation + lmm
- **KNN-lmm:** log-transformation + kNN imputation + lmm
- **SVD-lmm:** log-transformation + SVD imputation + lmm

## Hurdle model

*Goeminne et al. (2020)* proposed a peptide level model, the Hurdle model, that presents similarities with our approach, by combining MSqRob, a mixed model applied on all peptides observed intensities of each protein including a random sample effect, and a quasi-binomial model on the number of observed unique peptides. The *p*-values are combined as in (1), but independence is assumed between both statistics. Functions to implement the tests are available at https://github.com/statOmics/MSqRobHurdlePaper.

Following *Goeminne et al. (2020)*, additional filterings at the peptide level were applied to *ProteoCardis* and *Pigs* datasets prior to the Hurdle test implementation. Peptides with only one identification were deleted, as the model would be perfectly confounded. Then, proteins identified by only one peptide were removed.

## Resampling-based procedure for control of false discovery rate (FDR)

To account for correlation between variables, the false discovery rate (FDR) was controled using the resampling-based procedure proposed by *Reiner, Yekutieli & Benjamini (2003)*: *p*-values were reestimated by resampling (100 times) from the marginal distribution prior to *p*-value adjustment (*Benjamini & Hochberg, 1995*). Even if most FDR procedures only guarantee an upper-bound control and are subject to assumptions on dependence between variables, the number of selected variables for a given FDR threshold is an indicator of the FSM's power. Moreover, the resampling-based FDR procedure considered here enables the bias due to complex dependence structures to be circumvented. Mathematical details are provided in the Supplemental Material.

## Criteria for method comparison

### Agreement between FSMs

The overall similarity between FSMs on all proteins was measured by Kendall correlation between $p$-values which enables a non-parametric comparison, as well as Pearson correlation on log-transformed p-values, which gives more importance on consistency between small $p$-values than between large $p$-values (since the log transformation spreads the values close to 0 and packs down the values close to 1). The proportion of common selected features among the top $N$ directly targets agreement in terms of feature selection. Values of $N$ were chosen for each data set according to the number of significant features. For *ProteoCardis* data sets which display a small number of significant values, we considered $N = 30, 100, 200$. For *Pigs*, as a large number of proteins were significantly different between the two classes, we considered the larger values $N = 200, 500, 1,000$. Moreover, for *Pigs*, sample splitting in cross-validation loop and stability computations was implemented such that all observations from the same animal remained in the same subset.

### Stability of feature selection between independent data sets

The set of samples was repeatedly (100 times) split into two independent subsets while preserving the proportion of each class. FSMs were applied on each subset, and the stability was quantified as the proportion of common variables among the top $N$ features selected on each of the two subsets.

For comparison between the combined test and the Hurdle test, we considered alternative criteria that account for the difference in the total number of features, since the number of tested proteins differs due to the additional filtering for the Hurdle model. Cohen's kappa (*McHugh, 2012*) enables users to compare the agreement between two "raters" (here, the selection based on the two independent subsets) with the chance agreement. Fisher exact test and $\chi^2$ association test quantify, in a non-parametric and parametric way, respectively, the association between selections operated on the two subsets.

### Classification accuracy

Classification accuracy was computed on a 10-fold cross-validation loop for *ProteoCardis* repeated 10 times to evaluate the standard deviation and a leave-one-out procedure leaving out all observations from each animal in turn on *Pigs*. The classification procedure consists in selecting the top $N$ proteins, and then infer either a random forest (RF) or a support vector machine (SVM) based on these $N$ features. For prediction, the missing values were replaced by zero. Filtering was performed on the complete data set (as this step does not involve class labels). For each cross-validation split, the whole classification procedure including feature selection was performed on the training set only, then the labels of the samples in the validation set were predicted.

## RESULTS

### Missingness mechanisms: both MAR and MNAR

The replicate data sets, including seven technical replicates for eight biological samples, allow for the assessment of the technical variability and the analysis of the MAR/MNAR hypotheses. Figure 1 and Figure S2 (left) display the average observed intensity of a protein as a function of the number of times it is missing in the replicate samples. The observed intensity decreased as the number of missing values increased, which suggests that the probability to be missing is higher when the protein abundance is lower, so missingness mechanisms is at least partially MNAR. In particular, we observed a pronounced difference of intensity between proteins with no missing value and protein with one, or *a fortiori* more than one missing values. Nevertheless, even when the protein was missing in a large proportion of replicates, the average observed intensity could still be high, indicating that missingness mechanisms are not exclusively MNAR. These observations were confirmed by the probability of being missing, that decreased when the observed intensity increased, but remained non-negligible even for consistent observed intensities (Fig. 1 and Fig. S2). For example, for an intensity equal to the median of the observed values on the data set, the probability of being missing was 0.23–0.25, and even for an intensity equal to the 0.9 quantile, the probability was still 0.05.

### Comparison of FSM's performances on simulated data

Figure 2 illustrates the ability to discriminate differentially and non-differentially expressed proteins of three FSMs, assuming two missingness mechanisms (MAR and MNAR) and under two types of difference of expression between proteins (differential presence and differential abundance). We considered two imputation-based FSMs: SVD based on global structure similarity, and single value imputation by the smallest observed intensity, followed by a linear (mixed) model, as well as the combined test. kNN-lmm was not considered since kNN imputation was used to generate the simulated data set. FSM's performances were highly impacted by both the nature of the difference of expression and the missingness mechanism. Each imputation based FSM failed under one scenario: single-lmm did not achieve to detect differentially abundant proteins under MAR, while the AUC with SVD-lmm was close to 0.5 when proteins were differentially present under MNAR assumption. These observations are coherent with the underlying assumptions under each imputation methods: MAR for SVD and MNAR for single value imputation. In all scenarios, the combined test remains competitive, with AUCs close to the ones of the best of the two imputation-based FSMs.

### Poor concordance between imputation-based FSMs based on MAR and MNAR

Comparison of the three imputation-based FSMs in terms of correlation between log-transformed *p*-values and proportion of common selected features (Fig. 3 and Fig. S3) indicated a strong agreement between SVD-lmm and KNN-lmm, but both methods showed a poor concordance with single-lmm. These observations were common to the

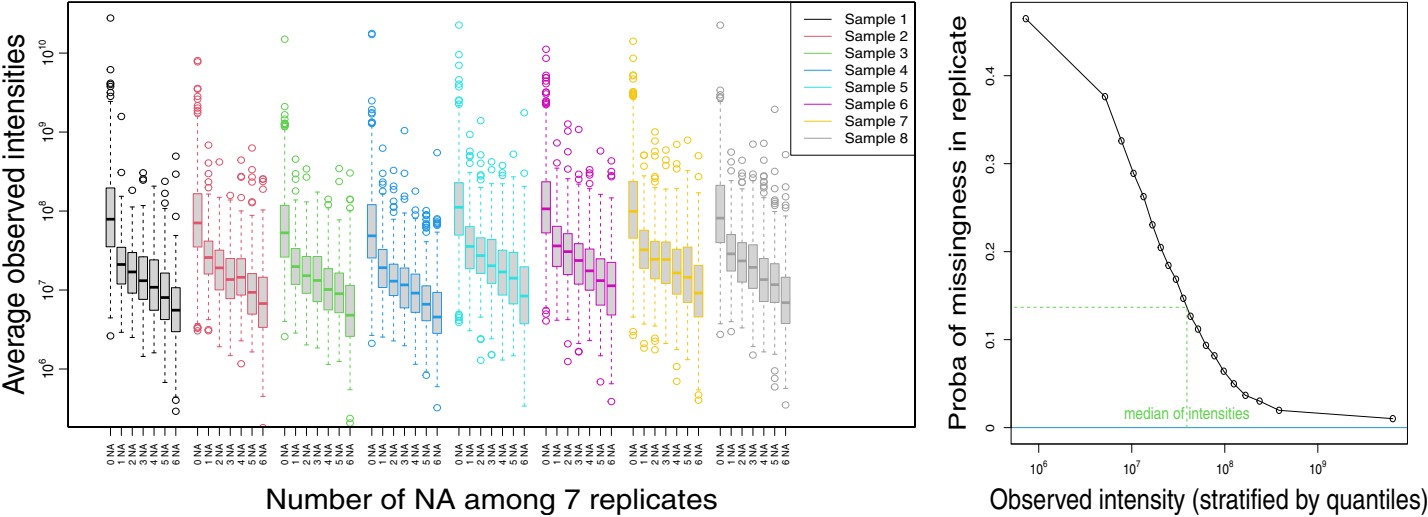

**Figure 1 Analysis of replicates-cytosolic fraction.** Left: log10-transformed average intensities of non-missing observations as a function of the number of missing values (NA), for all proteins and for each biological sample. Right: estimate of the probability that a protein is missing when the analysis is replicated, as a function of the average of its non-missing values.

*ProteoCardis* and *Pigs* data sets, but the concordance between methods was globally higher for *Pigs* due to a weaker proportion of missing values, which reduced the impact of the imputation method.

## Imputation-based FSMs are concordant with either glmm on probability of missingness or lmm on observed intensities

In addition, the imputation-based FSM *p*-values were compared with the two tests involved in the combined test (Rows 2 and 3 of Figs. 4, 5 and Fig. S4). We observed a very strong correlation between the *p*-values of mod-glmm and single-lmm. Indeed, as the smallest intensity used for imputation was far from most of the observed intensities (Fig. S9), the proportion of imputed values among a class strongly impacted the average intensity after single value imputation (a large proportion of missing values automatically leads to a small average intensity after single value imputation). Therefore, testing the difference in fold-change and in missingness leads to consistent *p*-values. Single-lmm *p*-values were weakly consistent with mod-lmm on observed intensities. Conversely, the *p*-values from KNN-lmm and SVD-lmm correlated well with mod-lmm, but weakly with mod-glmm.

## The combined test reaches a compromise between imputation-based FSMs

The combined test displays a strong agreement with single-lmm and a moderate agreement with KNN-lmm and SVD-lmm in terms of correlation between log-transformed *p*-values and proportion of common selected features (Fig. 3 and Fig. S3). This observation was confirmed by the scatterplots between log-transformed *p*-values of the combined test and each imputation-based FSM (first row of Figs. 4, 5 and Fig. S4). Indeed, features found highly significant by any of the imputation based FSMs were at least moderately significant with the combined test, while the opposite was not true. More

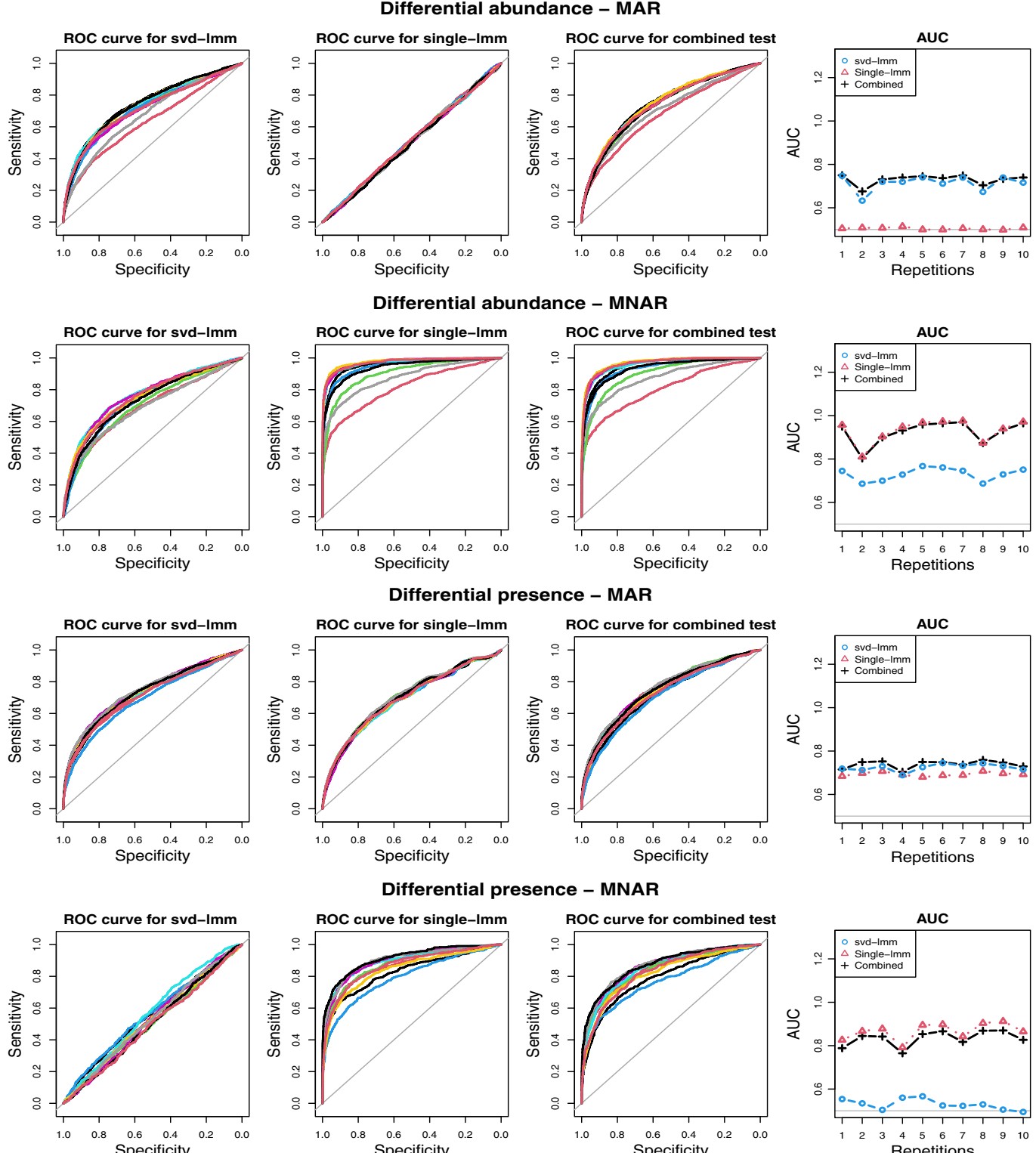

**Figure 2 ROC curves for various scenarios on simulated data.** Two types of difference between classes are generated: differential abundance of non-missing values and difference in probability of absence, and two missingness mechanisms: Missing At Random (MAR) and Missing Not At Random (MNAR). For each scenario, the two classes and the proteins that differs between classes are randomly sampled 10 times, and the ROC curve with the three FSMs: linear model after SVD (svd-lmm) and single value imputation (single-lmm), and the combined test (each color corresponds to a repetition). Column 4 displays the area under the curve (AUC) for each method for the 10 repetitions.

## ProteoCardis-cyto

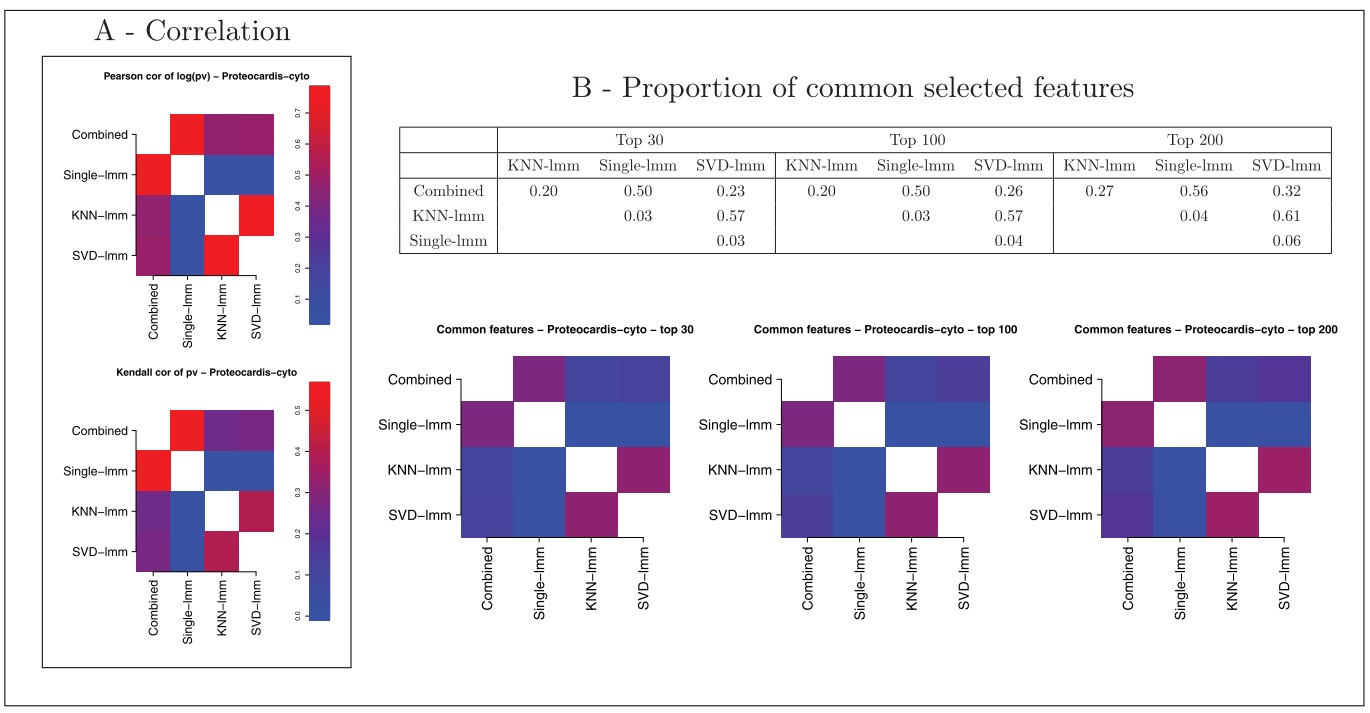

## Pigs

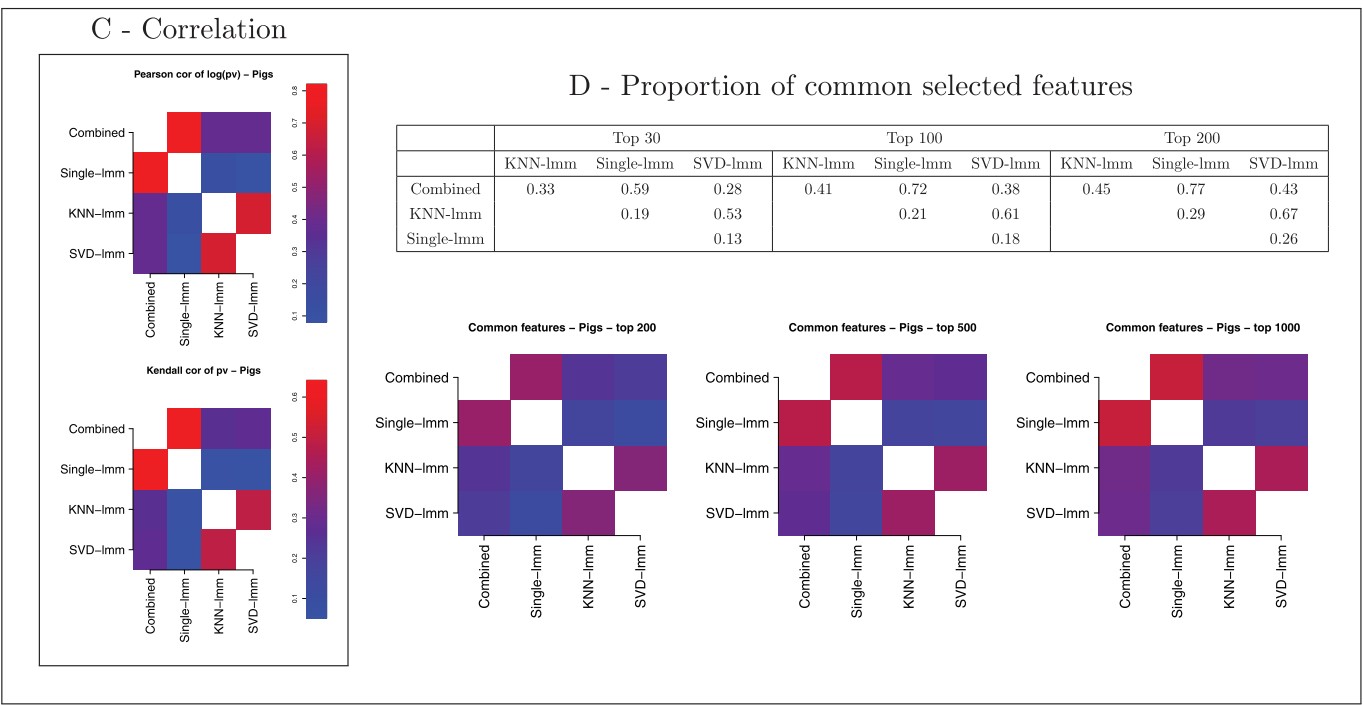

**Figure 3** **Pairwise agreement between *p*-values of FSMs.** (A, B) *ProteoCardis-cyto*; (C, D) *Pigs*. (A, C) Pearson correlation between log-transformed p-values and Kendall correlation between *p*-values. (B, D) Proportion of common features among the top *N* for each pair of FSMs, as a table and a heatmap.

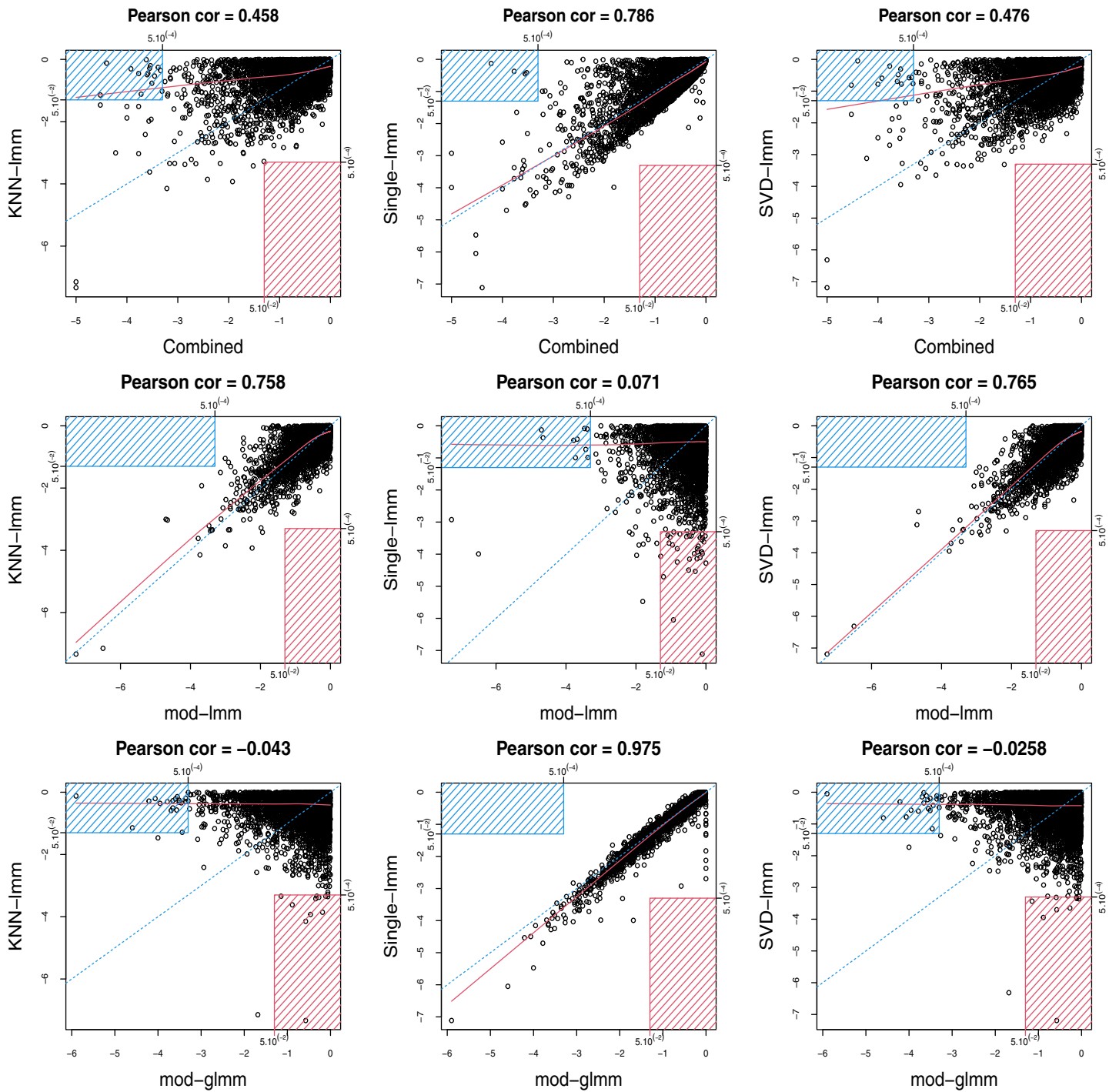

**Figure 4 Scatterplots between log10-transformed *p*-values of pairs of FSMs for *ProteoCardis-cyto*.** Row 1: combined test and imputation-based FSMs. Row 2: linear mixed model on observed values and imputation-based FSMs. Row 3: generalised mixed model (logistic) on missingness and imputation-based FSMs; proteins with less than 2 non-missing values are not displayed. For each pair of testing procedure, the red rectangle corresponds to proteins with $p > 5.10^{-2}$ with the first procedure and with $p < 5.10^{-4}$ for the second procedure; conversely, the blue rectangle corresponds to proteins with $p < 5.10^{-4}$ with the first procedure and with $p > 5.10^{-2}$ for the second procedure. Blue dotted line corresponds to the axis y=x, and red line to the lowess interpolation.

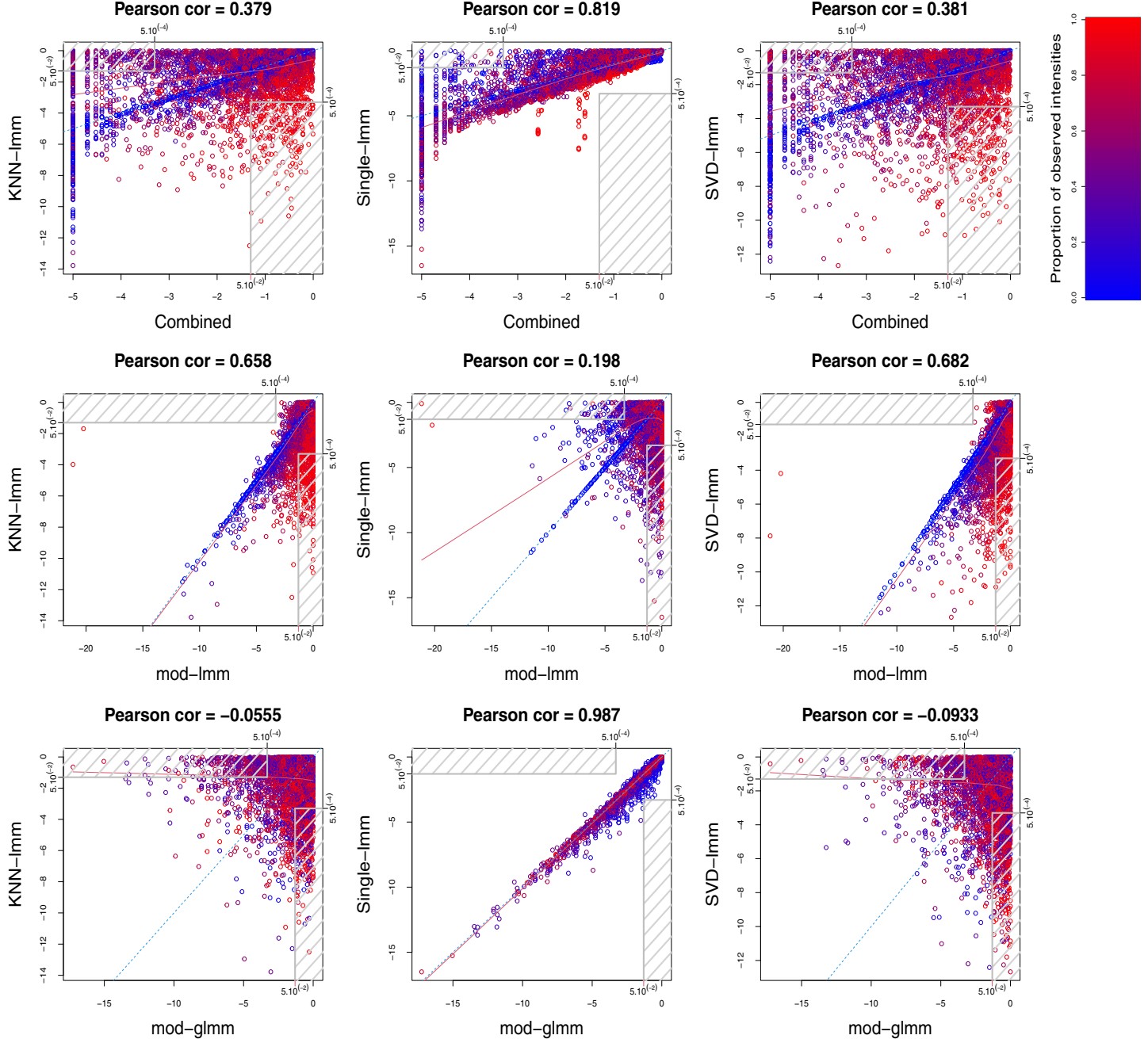

**Figure 5  Scatterplots between log10-transformed *p*-values of pairs of FSMs for *Pigs*.** Row 1: combined test and imputation-based FSMs. Row 2: linear mixed model on observed values and imputation-based FSMs. Row 3: generalised mixed model (logistic) on missingness and imputation-based FSMs; proteins with less than 2 non-missing values are not displayed. Color gradient corresponds to the proportion of non-missing values for each protein. Gray rectangles correspond to features with $p < 5.10^{-4}$ with one FSM and $p > 5.10^{-2}$ with the other. Blue dotted line corresponds to the axis y=x, and red line to the lowess interpolation.               

precisely, for *ProteoCardis* data sets, the proteins with very low *p*-values ($p < 5.10^{-4}$) with the imputation-based FSMs also had low *p*-values with the combined test ($p < 5.10^{-2}$); Conversely for each imputation-based FSM, some non-significant proteins displayed very low *p*-values with the combined test. A more nuanced but similar assessment holds on *Pigs*,

since most of the proteins that were found significant with SVD-lmm or KNN-lmm and non-significant with the combined test were very sparse (Fig. S5) and thus included a large proportion of imputed values, which indicates that imputation-based analysis on these variables is weakly reliable. On the contrary, the variables significant with the combined test but non-significant with SVD-lmm and KNN-lmm have a sparsity level that is either low or high.

## Impact of the filtering threshold

The impact of the imputation method is expected to decrease with the proportion of missing values, which itself depends on the filtering threshold. Therefore, we repeated our analyses with higher filtering thresholds: 30, 40 and 50 for *ProteoCardis*, and 20 and 30 for *Pigs*, to examine to what extent the comparisons between FSMs were impacted. Figures S6–S8 display the concordance between FSMs for several filtering thresholds.

The comparisons between methods still holds when threshold varies: SVD-lmm/KNN-lmm on the one hand, and single-lmm/combined test on the other hand were strongly concordant, while KNN-lmm/SVD-lmm were moderately concordant with the combined test, and poorly concordant with single-lmm. As expected, the agreement between all pairs of FSMs globally increased with the threshold.

A harder thresholding also resulted in a higher number of discarded variables, potentially leading to a loss of biological information. Indeed, Fig. S10 displays the distribution of the *p*-values as a function of the protein sparsity; we observed that very sparse proteins may exhibit very small *p*-values, in particular for *ProteoCardis* data sets. Moreoever, Table S1 indicates that the sets of most significant proteins for each FSM include 60–82% (average 70%) of features with more than half of missing values for *ProteoCardis*, and 17–57% (average 41%) for *Pigs*. Note that this proportion of sparse proteins among the selected ones is only slightly lower than in the entire data sets (79% for *ProteoCardis* and 43% for *Pigs*). This indicates that sparse proteins are selected almost as frequently as less sparse ones.

## Similar FSM's quantitative performances

Table 1 and Table S2 display the prediction accuracy with SVM and RF classifiers applied on selected proteins. Even though the four FSMs selected very different sets of features, their performances in terms of prediction were not significantly different.

Figure 6 and Figure S11 display the concordance between variable selection performed on independent data sets. On *ProteoCardis-cyto*, reproducibility of feature selection was similar with the four FSMs, while on *ProteoCardis-env* and *Pigs* the combined test and single-lmm outperformed imputation-based FSMs. Therefore variable selection based on the combined test and the single imputation FSM were equally reproducible and tended to be more reproducible than FSMs using structure-based imputation.

Figure S1 displays the number of selected variables for various values of FDR. The methods ranking varied with the data set and the FDR threshold, but the combined test remained competitive in terms of number of selected variables.

**Table 1 Prediction accuracy for two classification procedures on *ProteoCardis-cyto* and *Pigs*.** The selection of the top $N$ variables was followed by SVM or RF. For *Proteocardis-cyto*, accuracy was computed in a 10-fold cross-validation loop, repeated 10 times. For *Pigs*, accuracy was computed in a leave-one-out setting in which training sets consist in all measurements from one pig. Each cell provides the average accuracy (standard deviation of accuracy) computed over the 10 repetitions of the cross-validation. Bold numbers correspond to the highest accuracy among the four FSMs.

| | | Combined | KNN-lmm | SVD-lmm | Single-lmm | Hurdle |
|---|---|---|---|---|---|---|
| *ProteoCardis-cyto* | | | | | | |
| Top 30 | RF | **0.706** (0.0075) | 0.679 (0.011) | 0.649 (0.013) | 0.705 (0.024) | 0.689 (0.012) |
| | SVM | **0.718** (0.017) | 0.56 (0.013) | 0.606 (0) | 0.664 (0.022) | 0.668 (0.0032) |
| Top 100 | RF | **0.714** (0.017) | 0.651 (0.0085) | 0.697 (0.017) | 0.703 (0.021) | 0.704 (0.014) |
| | SVM | **0.722** (0.02) | 0.535 (0.032) | 0.688 (0.029) | 0.689 (0.0064) | 0.684 (0.0096) |
| Top 200 | RF | 0.698 (0.019) | 0.672 (0.02) | 0.702 (0.014) | 0.697 (0.013) | **0.712** (0.017) |
| | SVM | 0.708(0.019) | 0.638 (0.0064) | **0.716** (0.035) | 0.696 (0.0032) | 0.702 (0.016) |
| *Pigs* | | | | | | |
| Top 200 | RF | 0.903 | 0.903 | **0.917** | 0.903 | 0.903 |
| | SVM | 0.833 | 0.861 | 0.889 | 0.875 | **0.917** |
| Top 500 | RF | **0.917** | **0.917** | 0.903 | 0.903 | 0.903 |
| | SVM | 0.847 | 0.833 | 0.875 | 0.847 | **0.917** |
| Top 1,000 | RF | **0.917** | 0.889 | 0.903 | **0.917** | 0.903 |
| | SVM | 0.792 | 0.819 | 0.819 | 0.792 | **0.917** |

## Comparison of the combined test with the peptide-level hurdle model

Similarly to the combined test, the Hurdle model proposed by *Goeminne et al. (2020)* targets simultaneously difference of intensity on observed values and difference in probability of detection, but with a peptide level model. As expected, both procedures lead to consistent *p*-values (Fig. 7), with Pearson correlation of log-transformed *p*-values of 0.637–0.786, but far from identical. In terms of feature selection, some proteins highly significant with one method exhibit non-significant *p*-values with the other, in particular for *Pigs*.

Regarding prediction, both procedures led to similar accuracy on *ProteoCardis* data sets, and on *Pigs* with RF, but the combined test had lower performances with SVM than RF, while the hurdle test maintained a similar accuracy (Table 1 and Table S2).
No significant differences in terms of replicability were observed between the protein-level combined test and the peptide-level hurdle test (Fig. 8 and Fig. S12). More precisely, feature selection with the combined test was slightly more replicable than the hurdle test on *Pigs*, notably for the largest number of selected features, while the opposite was observed on the *ProteoCardis* data sets.

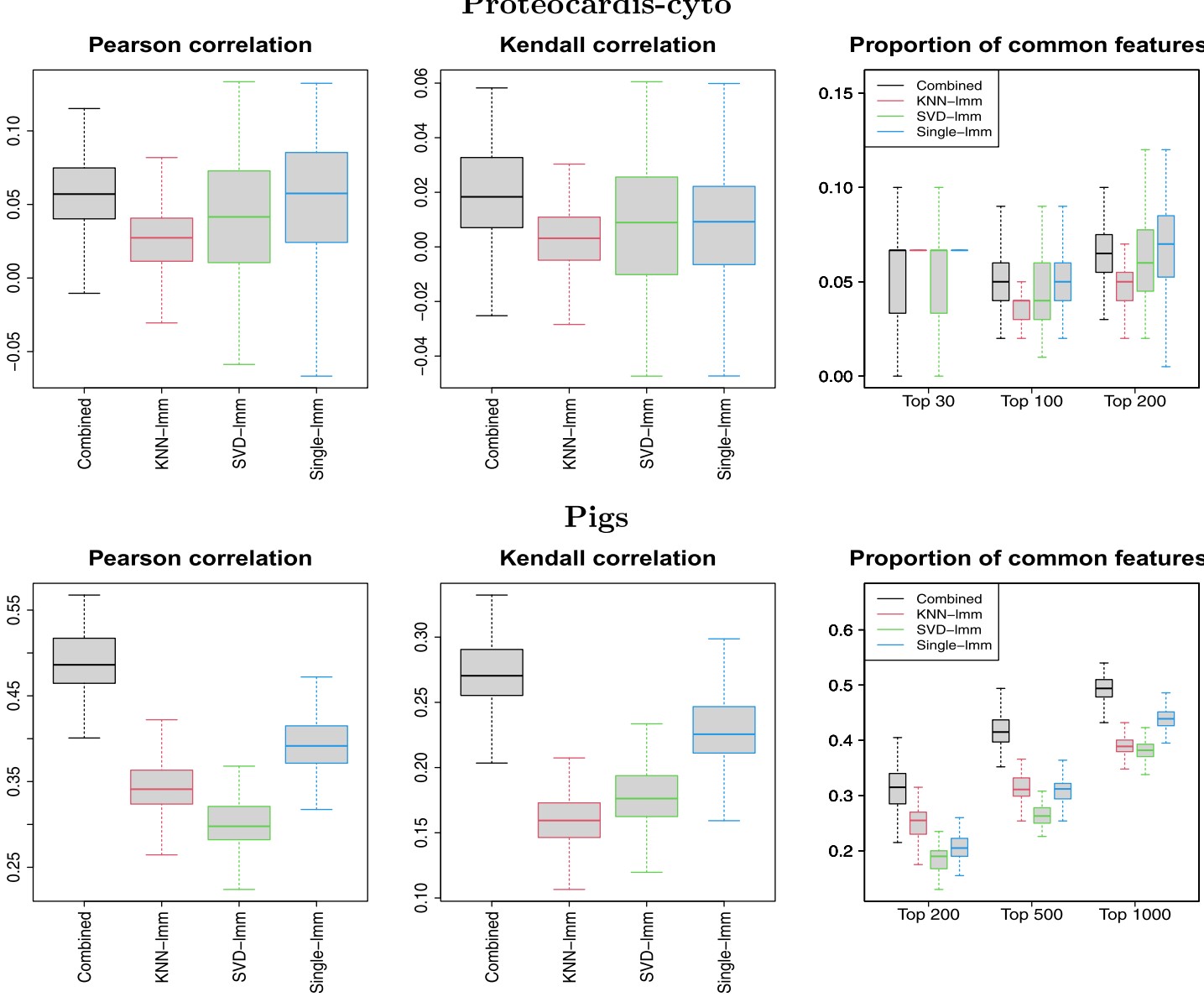

**Figure 6 Replicability of variable selection on independent subsets.** Pearson correlation between log-transformed *p*-values, Kendall correlation between *p*-values, and proportion of common selected features among the top *N*, for 100 splitting of samples into two subsets. Datasets: *Proteo-Cardis-cyto* and *Pigs*.

## DISCUSSION

### Missingness blends MAR and MNAR mechanisms and our method addresses both assumptions

Metaproteomics by LC-MS/MS generates a large proportion of missing values, usually imputed prior to statistical analysis. Several categories of imputation methods are routinely considered. Methods based on local similarity (*e.g.*, kNN) or global structure (*e.g.*, SVD) implicitly assume that missingness occurs independently of the true feature concentration (MAR). But the analysis of the replicate data sets clearly indicates that missingness is more

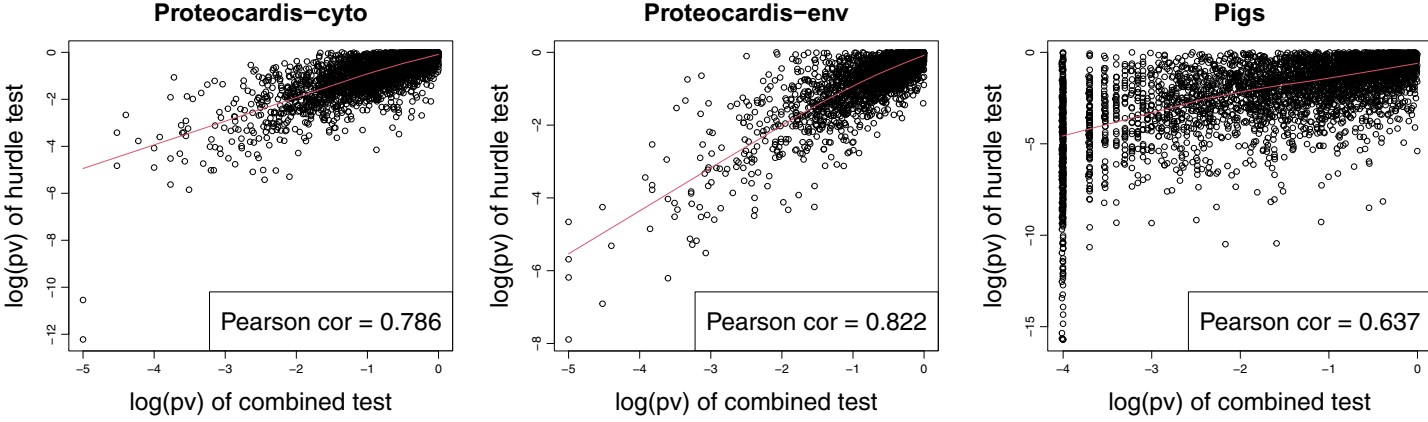

**Figure 7** Scatterplot of log10-transformed *p*-values of the hurdle test and the combined test for *ProteoCardis-cyto Protoecardis-env* and *Pigs* data sets.

likely to occur when the feature has a low abundance. On the other hand, the single imputation method relies on the assumption of a left censoring mechanism, but the distribution of observed intensities as well as the analysis of the replicate data set are not consistent with this assumption. Even if the co-existence of MAR and MNAR mechanisms is admitted in LC-MS/MS proteomics, exploration of their prevalence is often based on biological replicates (*e.g.*, *Karpievitch et al. (2009)*), assuming similar protein abundances in all samples from a given class. This assumption is questionable, in particular in human gut metaproteomics characterised by a strong individual specificity. On the contrary, our analysis based on technical replicates guarantees that the true protein abundances are identical.

## Limits of missing value imputation in metaproteomics

Missing value imputation is the standard method to account for missing values in metaproteomics. This flexible approach enables to address any type of statistical questions (*e.g.*, prediction, network inference,…) using methods developed for data with no missing values. But the downside is the risk to "forget" which values were imputed and to treat them equally to observed values, regardless of implicit assumptions underlying imputation that can strongly impact biological findings when a large proportion of values are missing (*O'Brien et al., 2018*; *Karpievitch, Dabney & Smith, 2012*; *Lazar et al., 2016*). In particular, we observed that global and local structure imputations specifically led to selection of features with a large proportion of imputed and thus weakly reliable values. Therefore, despite its easiness of use, imputation has a cost in terms of reliability and should be limited to moderately sparse data sets. On sparse metaproteomics data, this condition would require to filter out a large part of the features, which may be harmful since we demonstrated that a large part of the potentially interesting proteins have more than half of missing values.

As an alternative to missing value imputation, censored statistical models developed for proteomics data can account simultaneously for MAR and MNAR mechanisms (*Karpievitch et al., 2009*; *Luo et al., 2009*; *O'Brien et al., 2018*). Moreover, *Berg et al. (2019)*

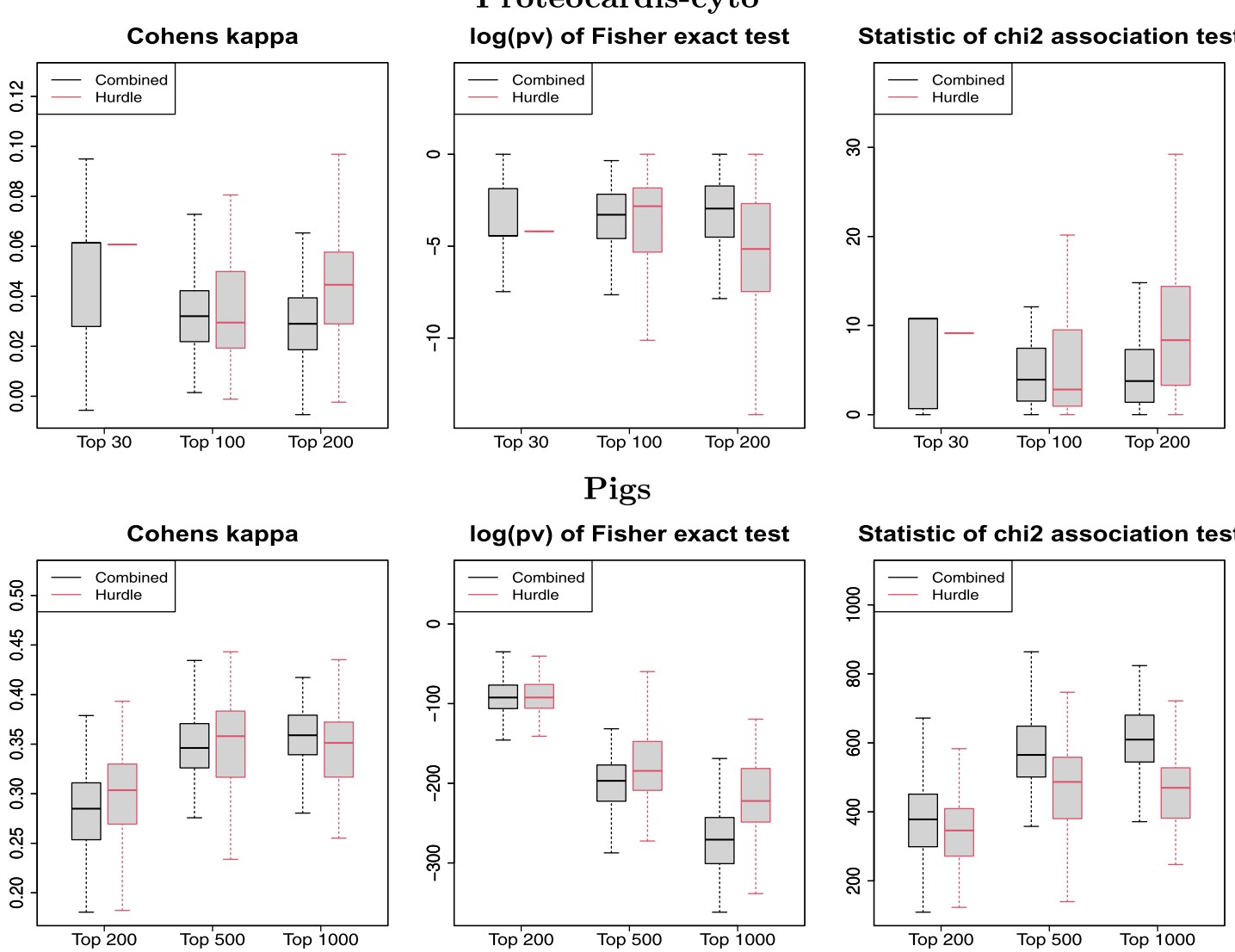

**Figure 8 Replicability of variable selection on independent subsets for the hurdle test and the combined test.** Boxplot of the Cohen's kappa (left), the log-transformed $p$-value of Fisher test (center) and the statistic of the $\chi^2$ contingency table test (right), for selection of the top $N$ features, performed on 100 splitting of the samples into two subsets. Black and red boxlots correspond to feature selection with the combined and the hurdle test respectively. Datasets: *ProteoCardis-cyto* and *Pigs*.

proposed a multiple imputation model which handles MAR and MNAR assumptions, but their method suffers from a methodological bias since imputation is performed within each class, which may artificially increase significance of inter-classes differences by enhancing intra-class similarities. Similarly, *Gianetto et al. (2020)* proposed a combination of MAR and MNAR imputation for proteomics data, but their method relies on a log-normal distribution of the true abundances, unrealistic in the metaproteomics framework where a large part of proteins are truly missing. Even though these models may address the complexity of missingness mechanisms more acutely than simple imputation methods, they also heavily rely on assumptions regarding missingness mechanisms (*e.g.*, hard thresholding) as well as signal distribution (*e.g.*, additive effect, Gaussian

distribution). Therefore, they can not be directly applied to metaproteomics data whose structure and characteristics strongly differ from proteomics.

## The combined test reaches a compromise between imputation-based methods

Beyond differences in underlying assumptions, distinct imputation-based FSMs lead to very different sets of selected variables. While selection with the two MAR imputation methods (SVD and kNN) were consistent, these two methods displayed almost no agreement with the FSM based on MNAR assumption (single value imputation) for the highly sparse data sets *ProteoCardis*, and a low agreement for the moderately sparse data set *Pigs*. Therefore, the choice of an imputation method can strongly impact the biological conclusions. The combined test, which addresses the two types of missingness by combining a glmm on probability of presence (relevant under MNAR assumption) and a lmm on observed intensities after removal of missing values (amounting to assume MAR mechanisms) displays a correct agreement with each imputation-based FSM. In greater details, the features detected using single value imputation were recovered by the glmm, while the features selected using kNN or SVD were recovered by the lmm on observed values, which is consistent with the assumptions on missingness mechanisms.

Prediction accuracy is a classic criterion to compare FSMs (*Tang et al., 2020b*), but its relevance is questionable when the main interest is the biological interpretation of selected features. Indeed, as enlightened in our analysis, methods with similar classification abilities can lead to totally different selected sets and feature ranking, so choosing a FSM based on a slightly higher prediction accuracy that - may vary with the chosen classifier - seems hazardous.

Furthermore, the simulation study conducted on archetypical scenarios illustrated that the comparative performances in terms of discrimination between differentially and non-differentially expressed proteins strongly depends on the nature of the protein difference (associated to the biological context) and the missingness mechanisms (mostly dependent on the technology). First of all, this simulation study indicates that quantitative comparison of FSMs' performances in a specific context has to be taken cautiously in another biological context, in which the proportion of each scenario, notably regarding biological differences, could vary. In particular, transposition of methods comparisons performed on proteomics data to the metaproteomics context can be hazardous, as both the biological context and the technical artefacts can differ. Secondly, while each imputation-based FSM fails in at least one scenario, the combined test remains efficient in all scenarios.

Thus, the combined test realises a compromise in terms of feature ranking and selection between inconsistent methods whose performances are based on unverifiable assumptions, and remains efficient in diverse scenarios. Therefore, it represents a robust solution, notably in the context of few prior knowledge regarding missingness mechanisms and the nature of biological differences.

### Protein *vs* peptide level analysis

In proteomics, analyses can either be realised at the protein or peptide level. In univariate testing procedures on proteomics data, peptide level analyses including a run effect are usually considered as more robust and with higher power (*Clough et al., 2012*). Nevertheless, in metaproteomics, analysis are routinely performed at the protein level (see review by *Tang et al. (2020b)*). This choice enables simplifications in the analysis, but biological arguments could be considered: the complexity of metaproteome may lead to a higher proportion of peptide misidentifications; moreover, the individual specificity of the metaproteome generates a very high sparsity at the peptide level (*e.g.*, 95–97% of missing values at the peptide level in the Proteocardis data sets) which is detrimental to the robustness of mixed model.

We compared the combined test with the hurdle model by *Goeminne et al. (2020)*, which displays similarities but is implemented at the peptide level. Interestingly, contrary to what was demonstrated by the authors on proteomics data, this peptide level analysis did not demonstrate superior performances compared to the protein-level combined test on our metaproteomics data sets. Further analyses of the biological findings brought by each method would enrich the comparison.

Beyond metaproteomics, the generic aspect of the combined test enables to use it in other omics or non-omics contexts with data missing both at random and not at random.

## CONCLUSION AND PERSPECTIVES

Feature selection based on imputation is highly dependent on the chosen imputation method, and thus on restrictive assumptions regarding missingness mechanisms, while biometric measurements typically subject to mixed missingness processes. Moreover, beyond censoring mechanisms, we enhanced the impact of different types of expression on FSMs' in performances. On the contrary, the combined test handles simultaneously missingness at random and not at random, and our analysis on metaproteomics data confirm its effectiveness to recover the strongest findings from imputation-based FSM based on either type of mechanisms and for different nature of biological changes, while displaying equivalent quantitative performances.

In this article, we focused on the missing data issue and we restricted our analysis to FSMs based on a linear mixed model, but the combined test could further be compared with more diverse FSMs, including wrapped and embedded methods (*Tang et al., 2020b*), as well as using data sets whose design include more than two classes. On the biological side, the conclusion of this article could be reinforced through validation by targeted proteomics measurements on a subset of variables. Ground truth data sets such as spike-in, where the concentration of a small number of features is controlled could also be considered, but one should keep in mind that comparative analyses of FSMs are strongly impacted by the type of biological differences between classes (notably differential presence or abundance). Finally, the combined test developed in this article is not restricted to metaproteomics data and could be implemented on other meta-omics data or on any data including a large part of missing values, whatever the missingness mechanisms. Moreover, the proposed approach could be generalised to univariate feature

selection in other frameworks than multi-class comparison (*e.g.*, time series) provided that a test of presence/absence is available (*e.g.*, a rank test for time series).

## ACRONYMS

| | |
|---|---|
| **FSM** | Feature Selection Method |
| **FDR** | False Discovery Rate |
| **kNN** | k-Nearest Neighbors |
| **SVD** | Singular Value Decomposition |
| **SVM** | Support Vector Machine |
| **RF** | Random Forest |
| **MAR** | Missing At Random |
| **MNAR** | Missing Not At Random |

## ACKNOWLEDGEMENTS

The proteomics analyses were performed on the PAPPSO facility (http://pappso.inrae.fr) which is supported by INRAE (http://www.inrae.fr), the Ile-de-France regional council (https://www.iledefrance.fr/education-recherche), IBiSA (https://www.ibisa.net) and CNRS (http://www.cnrs.fr). We thank Alfred AMEADAN (Univ. Paris-Saclay, INRAE, PAPPSO-Micalis, 78350, Jouy-en-Josas, France) for preparing, analysing, and generating the raw LC-MS/MS data sets of the ProteoCardis study. Sylvie HUET (senior research scientist at MaIAGE-INRAE, Université Paris-Saclay, 78350 JOUY-en-JOSAS), Christine CARAPITO (senior research scientist at IPHC-CNRS, 67037 STRASBOURG) and Magali ROMPAIS (research engineer at LSMBO, IPHC-CNRS, 67037 STRASBOURG) are gratefully acknowledged for their support and valuable contribution to scientific discussions.

### Funding

This work was supported by the Agence Nationale de la Recherche (ANR) as part of the ProteoCardis (ANR-15-CE14-0013) project. The funders had no role in study design, data collection and analysis, decision to publish, or preparation of the manuscript.

### Grant Disclosures

The following grant information was disclosed by the authors:
Agence Nationale de la Recherche: ANR-15-CE14-0013.

### Competing Interests

The authors declare that they have no competing interests.

## Author Contributions

- Sandra Plancade conceived and designed the experiments, performed the experiments, analyzed the data, prepared figures and/or tables, authored or reviewed drafts of the article, and approved the final draft.
- Magali Berland conceived and designed the experiments, performed the experiments, analyzed the data, prepared figures and/or tables, authored or reviewed drafts of the article, and approved the final draft.
- Mélisande Blein-Nicolas analyzed the data, authored or reviewed drafts of the article, and approved the final draft.
- Olivier Langella performed the experiments, authored or reviewed drafts of the article, and approved the final draft.
- Ariane Bassignani performed the experiments, authored or reviewed drafts of the article, and approved the final draft.
- Catherine Juste conceived and designed the experiments, performed the experiments, authored or reviewed drafts of the article, and approved the final draft.

## Human Ethics

The following information was supplied relating to ethical approvals (*i.e.*, approving body and any reference numbers):

MetaCardis subjects were recruited between 2013 and 2015 in clinical institutions in France (Pitié-Salpêtrière Hospital, Center of Research for Clinical Nutrition (CRNH), Institute of Cardio-metabolism And Nutrition (ICAN)), Germany (Integrated Research and Treatment Center (IFB) Adiposity Diseases in Leipzig) and Denmark (Novo Nordisk Foundation Center for Basic Metabolic Research (NNFCBMR) in Copenhagen) for the European project MetaCardis. All subjects provided written informed consent and the study was conducted in accordance with the Helsinki Declaration and is registered in clinical trial https://clinicaltrials.gov/show/NCT02059538. The Ethics Committee of each participating country approved the clinical investigation. The study was approved by the Comite de Protection des Personnes (CPP) Ile de France III no. IDRCB2013-A00189-36.

## Data Availability

The software in the form of R code, and the ProteoCardis datasets are available at INRAE: Plancade, Sandra, 2021, "A combined test for feature selection on sparse metaproteomics data-Scripts and data", https://doi.org/10.15454/ZSREJA.

## Supplemental Information

Supplemental information for this article can be found online at http://dx.doi.org/10.7717/peerj.13525#supplemental-information.

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
