# Peer review of "A combined test for feature selection on sparse metaproteomics data—an alternative to missing value imputation"

_PeerJ, doi:10.7717/peerj.13525_

## Round 0.1 · original submission · Major Revisions

Dear Dr. Plancade and co-authors,

As you can see, our reviewers found that your study was important and interesting, however, they provided several comments and suggestions to strengthen your manuscript.

Reviewer 1 was critical and pointed out some experimental design issues. Reviewer 2 had a concern about the use of Pearson correlation. Reviewer 3 also had several comments. For example, it included the invalid statistics used on one dataset.

I agree with almost all the comments and suggestions. I also think that
improving the manuscript and in-depth checking your approach will
make your work stronger. I would like to ask you to address or to
respond with reasons not to follow the suggestion made by these
reviewers.


Best regards,
Atsushi Fukushima

Reviewer 1 ·

Basic reporting

Overall this article was clearly written although the organizations were not logical.
References and background are relatively clearly provided.

Experimental design

However, this study had some design issues. For example,
1. The proposed combined test was based on analysis of technical replicates.
The authors stated that this approach can guarantee that the true protein abundances are identical. This is not convincing. Actually in real study, biological variations are more important than technical variations and thus the methods that can detect biological variations are better than the methods based on technical replicates.

2. The combined Fisher combined statistic was defined as average a Fisher exact test and t-test of their p-values, which needs details on the algorithms.
3. It is true that feature selection based on imputation is highly dependent on the chosen imputation method. However, how the combined test addresses the MNAR
and MAR mechanisms lacked of details. The valuations based on concordance and Pearson correlation are not solid measures. More appropriate measures such as MCC or ROC are required.

4. The assumption that all variables display the same proportion of missing values is strong. Actually as the title suggested that metaproteomics data are sparse. Different samples typically have different rates of missing data. The combined method need to design different proportions of missing values to test its robustness.

5. More important, a simulation study is needed.

Validity of the findings

Further works are needed to confirm the findings and the conclusion is not robust.

·

Basic reporting

No comment.

Experimental design

No comment.

Validity of the findings

Handling missing data in proteomics is still an open major issue. Some works use single or multiple imputation to cope with that issue. The authors of this paper propose an alternative methodology for feature selection methods without imputation. The results provided by the authors are supported by various benchmarks and looks promising.
My only concern is about the use of Pearson correlation in several graphs (S3, S6, S7, S8, S11)? Why not use more general correlation coefficient such as Spearman or Kendall since Pearson correlation only detects linear links between variables?
Specific imputation methods were develop for proteomics featuring MEC and POV missing values patterns. For instance, the imp4p methodoogy by Quentin Giai Gianetto
(https://cran.r-project.org/web/packages/imp4p/index.html) tries to addresse them. Yet, it is not cited in the article and it would be interesting if it could be included in the benchmarks. The same suggestion fir the article Goeminne et al. 2020 (https://doi.org/10.1021/acs.analchem.9b04375)

Reviewer 3 ·

Basic reporting

no comment

Experimental design

no comment

Validity of the findings

See attached document for issues and necessary improvements.

Additional comments

See attached document for additional issues.

Annotated reviews are not available for download in order to protect the identity of reviewers who chose to remain anonymous.

---

## Round 0.2 · Minor Revisions

Dear Dr. Plancade and co-authors,

Thank you for your revision. Reviewer 2 still had minor comments about the revised manuscript. Especially, the calculation of correlations.

Would you please consider his/her comments to improve the manuscript?

Best regards

·

Basic reporting

No comment

Experimental design

No comment

Validity of the findings

Congratulations to the authors for having greatly improved the manuscript. The authors took into account some of my suggestions.

Yet, I still think that the use of Pearson Correlations should be replaced by Spearman's or Kendall's ones (Kendall's are generally recommended over Spearman's). Those two types of correlations can capture more general associations the Pearson's ones. They can be used to establish whether two variables may be regarded as statistically dependent. On the contrary, Pearson ones can be used to only establish whether two variables may be regarded as linearly statistically dependent. Moreover, Spearman's or Kendall's correlations can often be used as test statistics in more general settings than Pearson's ones. These tests are non-parametric, as they do not rely on any assumptions on the distributions of X or Y or the distribution of (X,Y). Hence, they may have been used to look for significant associations between the log-transformed p.

I am still wondering why would using Pearson's correlation implies: "Conversely, the order of low p-values has much more importance and this is what is captured by the Pearson correlation via the log-transformation."
On the contrary, it is known that Spearman's or Kendall's correlations capture the association of the variables up to a monotonic transformation and hence will more accurately capture the order of low p-values. With Pearson correlation, you can get a small correlation between variables that are dependent but not linearly (e.g. y=x^2). Why log transforming the p-values would imply getting linear associations between the logged variables? As can be seen on several graphs, it does not seem to be true.

As to the imp4p methodology by Quentin Giai Gianetto, it might have been adapted to zero-inflated data. Yet, I agree with the authors' statement "the imputation is realised separately for each biological condition - which is a way of proceeding that can artificially increases significance of inter-classes differences by enhancing intra-class similarities, and generate a bias in the p-values from subsequent testing procedures."
I am glad that you found the approach by Goemmine very relevant and that you added it to the benchmark.

There is one typo at line 480: UniversitÃ(c) -> Université.

Reviewer 3 ·

Basic reporting

No comment

Experimental design

No comment

Validity of the findings

No comment

Additional comments

I am believe that the authors have adequately addressed the comments from myself and other reviewers. The revised manuscript reads significantly better than the initial submission, and the authors have done a good job adding relevant details necessary for clarity and reproducibility. The addition of a simulation study has added substantial improvement to the manuscript.

---

## Round 0.3 · Minor Revisions

Dear authors,

Thank you for your revision. Would you please consider the comments provided by Reviewer 2?

Best regards

·

Basic reporting

no comment

Experimental design

no comment

Validity of the findings

I thank the authors for the additional details on the use of the correlation coefficients that they provide in this revised version of the article.

It leads me to a final comment on the use of a log transform and of Pearson's correlation coefficient.

In the setting of the article, logging data results in two different effects: changing the kind of functional relationship that may be detected using Pearson's correlation as well as weighting differently the domains on which the correlation coefficients are computed. Apparently, the latter effect is of high importance since the authors showed that Pearson's correlation coefficient used after a log transform has better properties than using directly a non parametric correlation coefficient on non logged values.

As a result, it would have been even more meaningful to compare Pearson's correlation coefficient and nonparametric correlation that were computed on the logged data. It would have enabled the authors to detect nonlinear relationships as well as look for significance.

Additional comments

no comment

---

## Round 0.4 · accepted · Accept

Dear authors,

Thank you for your revision.

Best regards

·

Basic reporting

no comment

Experimental design

no comment

Validity of the findings

no comment

Additional comments

I was also suggesting that using Kendall's correlation would have allowed searching for significance.
Nevermind, the article now addresses all my concerns.